# Coastal Impacts of Storm Gloria (January 2020) over the Northwestern Mediterranean

Angel Amores[1], Marta Marcos[1,2], Diego S. Carrió[3,4], and Lluís Gómez-Pujol[5]

[1]Mediterranean Institute for Advanced Studies (IMEDEA, UIB-CSIC), Esporles, Spain.
[2]Department of Physics, University of the Balearic Islands, Palma, Spain.
[3]School of Earth Sciences. The University of Melbourne, Parkville, Victoria, Australia
[4]ARC Centre of Excellence for Climate Extremes. The University of Melbourne, Parkville, Victoria, Australia
[5]Earth Sciences Research Group, Department of Biology, University of the Balearic Islands, Palma, Spain

**Correspondence:** A. Amores (angel.amores@uib.es)

**Abstract.** The ocean component and coastal impacts of Storm Gloria, that hit the Western Mediterranean between January $20^{th}$ and $23^{rd}$ 2020 are investigated with a numerical simulation of the storm surges and wind-waves. Storm Gloria caused severe damages and beat several historical records such as significant wave height or 24-h accumulated precipitation. The storm surge developed along the eastern coasts of the Iberian Peninsula reached values up to 1 m, and were accompanied by wind-waves

with significant wave height up to 8 m. Along the coasts of the Balearic Islands, the storm footprint was characterised by a negligible storm surge and the impacts were caused by large waves. The comparison to historical records reveals that Storm Gloria is one of the most intense among the events in the region during the last decades and that the waves direction was particularly unusual. Our simulation permits quantifying the role of the different forcings in generating the storm surge. Also, the high spatial grid resolution down to 30 m over the Ebro Delta, allows determining the extent of the flooding caused by the

storm surge. We also simulate the overtopping caused by high wind waves that affected a rocky coast of high cliffs ($\sim 15\,m$) in the eastern coast of Mallorca Island.

## 1 Introduction

On 17 January 2020, a low-pressure system coming from the Atlantic made landfall in the northwestern part of the Iberian Peninsula. During the subsequent days, this low-pressure system evolved towards the southeast until reaching the Western

Mediterranean Sea (Figure 1) on 19 January, where it intensified, severely affecting the northern and eastern regions of the Iberian Peninsula, including the Balearic Islands. This low-pressure system was named Gloria by the State Meteorological Agency (AEMET). However, the life-cycle of Gloria last approximately 24 h, since it was absorbed by a larger low-pressure system that was centred over the Alboran Sea and last until 26 January. For simplicity, in this study we refer to Storm Gloria as the most intense activity period of this low-pressure system which ranges between 20-23 January.

The northwestern and the central sectors of the Western Mediterranean are modulated during most of the year by northerly winds, resulting from gales developed over the Gulf of Genoa and the Gulf of Lions. These winds are the main source of sea storms along the Northwestern Mediterranean basin (Flamant et al., 2003) and generate wind-waves characterised by significant

wave heights ($H_s$) between 0.1 – 1m and peak periods ($T_p$) between 3-6 s (Gómez-Pujol et al., 2019). In contrast, during the event of Storm Gloria, the synoptic situation was dominated by a deep low-pressure system located in the southern part of the domain that generated strong easterly winds (Figure 2). The event was initially dominated by a strong and negatively tilted upper-level ridge located over the Eastern Atlantic, where record-breaking pressure values of 1050 hPa (highest pressure value registered by the MetOffice since 1957) were registered in the north-western Europe, more specifically over the British Islands, followed by an intense upper-level trough centred over the Iberian Peninsula (Figure 2a). This upper-level trough was associated with cold air aloft, favouring potential instability over the mainland and the western Mediterranean Sea. During the following days this trough suffered a disconnection from the main westerly jet stream and as a result an intense cut-off low pressure system was developed over the Iberian Peninsula, enhancing a south-easterly flow at mid- and upper-levels (Figure 2b). At low levels, the initial low-pressure system (Figure 2c) moved towards the edge of the cut-off low, favouring the deepening of such surface low-pressure system and thus, enhancing the easterly winds, bringing unstable air (i.e., warm and moist) towards the eastern part of the Peninsula and the Balearic Islands (Figure 2d). This fact, together with the prominent orography associated with these regions, acts as a triggering mechanism for convection, which in this case resulted in heavy precipitations and flash floods. In addition, the prolonged coupling of the strong easterly winds together with the sea surface, generated wind-waves affecting coastal regions in the entire basin. According with the data from Mahon buoy ($39.71°N$ - $4.42°E$), these high waves reached maximum height of 14.77 m (January $21^{st}$ 12:00UTC) in deep waters, whereas winds also caused storm surges of up to 70 cm along the Spanish mainland (see the online report of the Spanish Meteorological Agency AEMET at http://www.aemet.es/es/conocermas/borrascas/2019-2020/estudios_e_impactos/gloria, in Spanish only). As a consequence many coastal sectors were flooded, coastal infrastructures were destroyed and strong erosion was reported in sedimentary coasts, with immediate economic losses of several millions of euros (e.g. only in the region of Valencia economic losses are estimated in up to 62.6 million € to the agriculture sector, according to the Valencian Association of Agricultural Producers (AVA-ASAJA)). According to the Copernicus Emergency Management Service (EMS) (http://emergency.copernicus.eu) the damages caused by the storm included lost of power supply, flooded areas and coastal erosion, as well as a total of 13 fatalities (https://emergency.copernicus.eu/mapping/sites/default/files/files/EMSR422_Floods_in_Spain_0.pdf).

Our focus here is on the marine effects of the storm. We concentrate on the shorelines of the eastern Spanish coasts and the Balearic Islands, where major impacts were reported. To quantify the impacts of Storm Gloria we numerically simulate the storm surges and wind-waves generated over the Western Mediterranean Sea. We apply near-real time atmospheric forcing fields of atmospheric pressure and surface winds to a coupled hydrodynamic and wave propagation models. Our purpose is twofold: the first part has a regional scope in which we aim to quantify the physical mechanisms at play along the different coastal areas in the basin, including the storm surges and the effect of waves, and to discuss their differences that ultimately led to coastal impacts. The model outputs allow identifying the main drivers of coastal hazards for areas differently exposed to the effects of the storm. Secondly, we focus on local case studies. We select locations with different characteristics (morphological and in terms of forcing) and simulate the impacts of the storm, accounting for the storm surge and wave setup at the local scale. The first site is the Ebro Delta (Figure 1), a low-lying region (with elevation $\sim 1$ m) of 320 km$^2$, of which around 77 km$^2$ corresponds to a protected natural area, being one of the most important aquatic habitats in the Western Mediterranean. During

Storm Gloria, large parts of the delta were submerged. The second site corresponds to high-cliffs ($\sim 15$ m) in Portocolom, a small town located in the eastern side of Mallorca Island (Figure 1), where very high waves were reported (see picture in Figure 8) that overtopped the cliffs severely damaging coastal assets.

The manuscript is organised as follows. We describe the oceanic and atmospheric observations and the implementation of the numerical models and their atmospheric forcing in section 2. This section also includes the model validation (in subsection 2.3). This is followed by the analyses of the results of the regional modelling (section 3) and the two case studies: the first one describes the coastal inundation in the Ebro Delta induced by the storm surges and waves (subsection 3.1) and the second addresses the wave impacts in the rocky coast of the eastern Mallorca Island (subsection 3.2). The final section discusses the results and provides the final remarks.

## 2   Data and Methods

### 2.1   Ocean and atmospheric observations

Sea level observations from tide gauges, wave measurements from *in-situ* buoys and remote sensing altimetry have been used to describe and quantify the effects of the storm over the ocean and also to evaluate the performance of the numerical simulations. Wind and atmospheric pressure records have also been recovered from weather stations on the buoys and compared to the atmospheric forcing used to feed the ocean models.

Wave parameters including significant wave height ($H_s$), wave peak period ($T_p$) and wave peak direction have been retrieved from four deep-water buoys and two coastal buoys (see Figure 3). All buoys provide near-real time hourly measurements. These observations have been complemented with along-track measurements of $H_s$ from satellite altimetry on board of satellites SARAL/Altika, Cryosat-2, Sentinel 3A and B and Jason-3, with all missions homogenized with respect to the latter and calibrated on *in-situ* buoy measurements. All wave observations have been downloaded from the CMEMS data server.

Hourly tide gauge time series at five stations over the Northwestern Mediterranean basin are available in near-real time through the Copernicus Marine Environment Monitoring Service (CMEMS, http://marine.copernicus.eu). Two more tide gauges providing near-real time sea-level measurements every minute have been obtained from the Balearic Islands Coastal Ocean Observing and Forecasting System (SOCIB, https:www.socib.es). See Figure 4 for location of these tide gauge stations.

### 2.2   Numerical modelling of storm surges and wind-waves

The storm surge and wind-waves generated by Storm Gloria over the Western Mediterranean have been simulated using SCHISM model (Semi-implicit Cross-scale Hydroscience Integrated System Model; Zhang et al. (2016)). We have used its dynamic core, which is a derivative product built from the original SELFE (v3.1dc; Zhang and Baptista (2008)), in 2DH barotropic mode fully coupled with the spectral wave model WWM-III (Roland et al., 2012). The two modules share the same triangular unstructured computational grid that covers the whole Western Mediterranean basin (Figure 1) with a total of 390113 nodes distributed in 771297 elements. Its horizontal grid resolution ranges from $\sim$15 km in open ocean down to 1-2

km along most coastlines and $\sim 30$ m at the Ebro Delta. This small region covering the Delta and its surroundings contains around 75 % of the grid nodes. The Strait of Gibraltar, the region between Tunisia and Sicily and the Strait of Mesina were considered as closed boundaries. The model was implemented using the bathymetry 2018 version of the EMODnet digital terrain model (https://portal.emodnet-bathymetry.eu/), with a resolution of $1/16 \times 1/16$ arc minutes ($\sim 115\,m \times 115\,m$) over the European sea regions. In the Ebro Delta (upper-left panel in Figure 1) this bathymetry was combined with a high-resolution topography, provided by the Institut Cartogràfic de Catalunya, derived from LiDAR observations. The product covers an area of $24km \times 30km$ with a spatial grid resolution of 2 m. The modelled time period covered 9 days, from January $17^{th}$ to $26^{th}$ 2020. The computational time step was set to 10 minutes and the variables were saved every 30 minutes. The simulation took a total of 2 days and 19 hours and it run in a single core, what makes a performance of 3.2 times faster than the real time.

The atmospheric fields used to force SCHISM were retrieved from the high-resolution version of the deterministic ECMWF analysis (https://www.ecmwf.int/en/forecasts/datasets/set-i). These analyses are provided on a regular lat/lon grid and have the highest currently available ECMWF's horizontal grid resolution ($0.1° \times 0.1°$), which is roughly equivalent to 9 km in the mid-latitudes. In terms of temporal resolution, the outputs are available every 6 hours, i.e., at 0000 UTC, 0600 UTC, 1200 UTC and 1800 UTC. The atmospheric forcing terms correspond to the mean sea-level atmospheric pressure, the U-component and V-component of the wind at 10 m, spanning the period 6-27 January. We have compared point-wise modelled time series with measurements of the atmospheric parameters over the ocean at buoy locations (see the section above for details on the stations) in Figure S.M. 1. The results show good agreement at every location and for all the forcing terms, thus confirming the reliability of the atmospheric forcing fields. Tides were not considered in the simulation since the Western Mediterranean is a micro-tidal environment.

In order to evaluate the contribution of the different forcings factors, namely atmospheric pressure, wind and wave setup, to the total water levels along the coasts, four different simulations were designed: 1) a fully coupled run between the hydrodynamic model and the wind-wave module that takes into account all the forcings as well as their coupling; 2) a run without wind-waves, using only the hydrodynamic module forced by wind and atmospheric pressure; 3) a hydrodynamic model run forced only by atmospheric pressure; and 4) a hydrodynamic model run forced only by wind. The total storm surge generated by Storm Gloria was evaluated using only the simulation #1; the effect of the wind-waves and the coupling with the dynamic effects was estimated by the difference between run #1 and #2; finally, the contribution of the atmospheric pressure (wind) was determined with the run #3 (#4). To quantitatively determine the individual effect as well as the synergistic effect between a set of different factors, a total of ($2^{\#forcings}$) simulations are required (Stein and Alpert, 1993), with all the possible combinations of switching on and off all the forcing terms. However, if the difference between the fully coupled run (#1) and the combination of the different forcings $[(\#1 - \#2) + \#3 + \#4]$ is computed, the median relative difference emerging from the simulations is around 0% and the maximum relative difference in one coastal point is around 3%. These small differences indicate that the interaction between the different forcings is small and justifies not to perform the additional numerical simulations. Note, that even in absence of strong interaction between storm surge waves and wind-waves, the coupled simulation is needed to account for the effects of the wave setup along the coasts.

To explore the impacts of Storm Gloria at the local scale, we focus on two cases under different forcing conditions. In the first one, we quantify the flooding induced by the joint effect of storm surges and waves on the Ebro Delta, taking advantage of the local very high grid resolution that allows an accurate representation of the local topographic features. In the second case, we evaluate the nearshore wave propagation and overtopping in Portocolom, located on the eastern coast of Mallorca Island (Figure 1 and upper panel in Figure 8), a rocky coast formed by high cliffs ($\sim 15\,$m) where numerous damages were reported by waves overtopping. Note that spectral wave models, such as WWM-III model used here, are unable to represent the overtopping generated by individual ocean waves; therefore, a set of 1D simulation were performed with the non-hydrostatic phase-resolving model SWASH (Zijlema et al. (2011); http://swash.sourceforge.net/) along a bathymetric profile with 1 m of horizontal resolution extracted from a nautical chart (1:5000) from the Spanish Army Hydrographic (IHM) Institute (the location of the profile is indicated by a red line in the upper panel in Figure 8; the profile is shown in the lower inset in the upper panel in Figure 8). The left side of the profile was artificially set as a downhill in order to account for all the water that overtops the cliff (upper inset in Figure 8). The Manning roughness coefficient was considered constant in all the domain with a value of 0.025. The wave forcing provided by the closest grid point of the regional model (extracted from simulation #1) was introduced in the eastern side of the 1D domain with a JONSWAP spectra. With this configuration, the simulation time was 60 minutes, with an initial computation time step of 0.05 s. Given that the comparison between modelled and observed $H_s$ at the nearest buoy (Mahon) suggests that the model underestimates the magnitude of the waves (see section 2.3), instead of using the time series of wave parameters of the closest point, a set of values for $H_s$ have been tested, ranging from 5 to 10 m, every 0.5 m, to determine the minimum significant wave height needed to have overtopping on the cliffs. To minimize the inherent stochasticity when generating the wave forcing with a JONSWAP spectra, a total of 100 simulations with different seeds to generate the spectra were run for each $H_s$ analysed.

## 2.3  Model validation

Modelled storm surges and waves have been compared to observations for the period 17-26 January 2020. The time series of the closest model grid point (the distance between model grid point and observation is indicated in Figure 3) was extracted for comparison with *in-situ* measurements. For the comparison to altimetry data, the closest model grid point in space, but also in time, to each altimetry track was used. Satellite observations with distance larger than 10 km from the closest grid point or with a time difference larger than 15 minutes between modelled time and observed time were discarded.

Figure 3 displays the comparison between modelled and observed wave parameters. *In-situ* measurements of $T_p$ and wave direction are very well captured by the model output at both deep-water and coastal wave buoys. During the storm, $T_p$ ranges between 10-12 s everywhere, with directions from the west ($\sim 90$ degrees). Changes in these parameters are correctly simulated during the entire period with maximum differences in $T_p$ of only 1 s. $H_s$ significantly increases during the storm reaching values of up to 8 m, according to observations. The model underestimates $H_s$ at all buoy locations, except in the two buoys near Tarragona (light blue curves in Figure 3). Differences between model and observations for $H_s$ are between 1 m in Valencia (dark blue line) and 2.5 m in Mahon (red line) during the peak of the storm. This underestimation is also found consistently in the comparison to altimetry. The scatter plot of along-track altimetric observations and the corresponding model results displays

a significant correlation of 0.73, although with a slope between observed and modelled $H_s$ of 0.90, indicating that the overall underestimation is about 10% of $H_s$ over the western Mediterranean basin. The possible causes of this underestimation include a poor energy transfer between the atmosphere and the ocean as simulated by the WWM-III wave model, a limited atmospheric forcing (either in terms of temporal or spatial resolution), and inaccurate bathymetry in some locations. To discard the role of the wave model in the $H_s$ underestimation, we repeated the wave simulation with the same computational grid using SWAN model, a different spectral wave model developed at Delft University of Technology (http://swanmodel.sourceforge.net/) which led to equivalent results. As discussed above in the previous section, Figure S.M. 1 shows a good agreement between observations of atmospheric pressure and winds and the model forcing, with only small underestimations in wind velocity during the peak of the storm. Thus, we conclude that the likely causes of the $H_s$ underestimation are related to limited bathymetry and small inaccuracies in the atmospheric forcing fields.

Coastal sea surface elevation is compared in Figure 4 to tide gauge observations. Time series are referred to mean sea level at each site. The sea level time series recorded by the tide gauges (represented in different colours in Figure 4 panels) have been detided (using the complete record of each station) and low-pass filtered using a Butterworth filter with a cutoff period of 30 minutes to remove the resonant modes of the harbors where the tide gauges are located. The filtered non-tidal residuals were compared to modelled outputs at grid points located within a radius of 2.5 km from the tide gauge. This approach seeks minimising the effect of local topographic features in the differences between modelled and observed records. We recall here that, since we run the coupled version of the model, sea surface elevation accounts for the storm surge and the effect of wave setup (every contributor is discussed and quantified below). Notably, observations indicate that sea levels have temporarily risen for up to 70 cm along the coasts in the mainland (sea level rises up 1 meter in Gandia prior to detiding and filtering - not shown). In contrast, values are much lower (around 20 cm) at tide gauges in the Balearic Islands. The similarities between modelled and observed extreme sea levels are remarkable at every tide gauge station: in Gandia, Sagunto and Valencia, the locations with larger increases, the model mimics both the intensity and the evolution of the sea surface elevation. Only in the tide gauge station in Tarragona, the model is found to significantly underestimate ($\sim 15\,\mathrm{cm}$) observed sea levels; for the rest of the tide gauge records, the model results follow the oscillations and their magnitudes observed by the instruments. The underestimation of ocean surface elevation observed in Tarragona is likely related to inaccurate bathymetry at this location, a factor to which the storm surges are more sensitive at coastal locations.

The overall good agreement between modelled and observed sea level, as measured by tide gauges, suggests that the simulation correctly captures the most relevant processes that are taking place during Storm Gloria. In particular, the contribution of the wave setup can be quantified by comparing observations to both coupled and uncoupled runs. This is shown for the tide gauge in Gandia in Figure S.M. 3. Not accounting for the wave setup (red lines) underestimates by around 20 cm the observed sea level (blue line), whereas including the effect of waves (grey lines) decreases this bias. Indeed, the closest grid point (thick grey line) mimics the amplitude of the observed storm surge when including the effect of waves. This indicates that the spatial resolution along the coast ($\sim$ 1-2 km at this site) is enough to represent the effects of the wave setup.

 **3   Storm surges and wind-waves generated by Storm Gloria**

The fully-coupled regional model (simulation #1) was run for the period 17-26 January 2020. The three days before the storm serve as spin up of the model to avoid starting the storm simulation from rest. The simulation is shown in S.M. Video 1, where frames are saved every 30 min. The video shows the evolution of the atmospheric pressure and wind fields (top figure) together with the ocean responses in sea surface elevation (middle figure) and waves (bottom figure). The intensity of wind fields increases rapidly on January 19 on the northern sector of the Western Mediterranean. This is followed by an immediate increase in $H_s$ in the same area and direction; also visible is the accumulation of water along the coastal mainland resulting from the storm surge. On January 20, strong easterlies are developed in the centre of the basin, that generate waves of $H_s$ around 5 m that reach the eastern coasts of the Balearic Islands. Contrary to the mainland, these waves hitting the Balearic Islands are not accompanied by significant storm surges. Note that the colorbars of both variables are saturated for representation purposes.

Maximum values of $H_s$ and sea surface elevation during the storm at every grid point are mapped in Figure 5. Largest $H_s$ are distributed in the area between the Balearic Islands and the mainland (Figure 5a), reaching values over 8 m in the region of Alicante and especially in the vicinity of Denia (see the map for locations). The high storm surge is concentrated along the coasts of the mainland, extending from the North at the Ebro Delta towards the South of the domain (Figure 5b) and with values larger than 40 cm. In contrast, around the islands the storm surge is negligible. This pattern is caused by the winds blowing towards the shallow waters along the mainland, as is clearly inferred in the video. Again, colorbars in both $H_s$ and storm surge are saturated; the largest value of $H_s$ and storm surges are around 7.5 m and 60 cm, respectively. Overall, the results of the simulation indicate that the coastal impacts in the mainland were originated by local wind-waves reaching the coastlines on top of an extremely large storm surge component; conversely, in the Eastern Mallorca island, the impacts were caused solely by the effect of waves travelling from the East.

The set of model runs described in section 2.2 allows the quantification of every forcing factor on total water levels. The contributions of atmospheric pressure, wind and wave setup are analysed separately in a coastal stripe along the mainland and represented in Figure 6. The coastal stripe is composed of sectors of size ∼5 km in latitude and ∼10 km in longitude, for which time series of total water levels and the three individual contributors corresponding to the grid points within each sector are grouped together. The values in Figure 6a correspond to the maximum total water levels in each sector. But to avoid any misrepresentation due to the limited bathymetry, the $90^{th}$ percentile of the values of all time series is used. The corresponding values of every contributor are then identified at the same time step as the maximum in total water level. The major contribution to the storm surge is caused by the wind forcing (Figure 6d); maximum values reach 40 cm in the surroundings of Denia and are around 30 cm northwards up to the Ebro Delta. These quantities represent a contribution of approximately 70% of the storm surges along the entire coast, with the exception of the southernmost part of the coastal stripe where the surge does not exceed 10 cm. In this area the main contributor is the atmospheric pressure (Figure 6c). The effect of the atmospheric pressure is in general negligible where the storm surge is large, reaching values of 10 cm at most. The wave setup, computed as the difference in total water levels between the coupled simulation (simulation #1) and the simulation without the wind-wave module (simulation #2), is quantified in Figure 6b. Values larger than 20 cm are found near Denia, where the wind effects are

also maximum, and in the northern part of the Ebro Delta. In these areas the wave setup accounts for up to 40-50% of the storm surge as a result of the combination of forcings and the shoreline geometry. In these two spots there was a large wind contribution to the total surge (Figure 6d) allowing for more energetic waves reaching the coast; at the same time, particularly high waves hit the coast in a direction that is completely perpendicular to the shoreline (see the video of the simulation in the S.M.) maximising the accumulation of water due to the effect of the waves. This is particularly critical in the low-lying region of the Delta, as shown and discussed in the next section.

## 3.1 Coastal flooding in the Ebro Delta

Shortly after Storm Gloria, Copernicus EMS produced and published an image based on Sentinel-1 satellite observations of the flooded area in the Ebro Delta, one of the regions most impacted by the storm (image available at http://floodlist.com/europe/ spain-storm-gloria-floods-january-2020). The satellite image maps the flooding during the days 20-22 January and shows the extensive and striking devastation over most of the Delta region. Note, however, that the image does not discriminate between coastal storm surge and rain-induced flooding, and that such discrimination is important to differentiate the parts flooded by salty water from those flooded by fresh water, as their impacts are very different especially for agricultural areas.

Our numerical simulation provides the total flooded area over the high-resolution Ebro Delta topography induced by the storm surge and waves. The results, mapped in Figure 7, show the maximum modelled flooding during Storm Gloria from January $17^{th}$ to $26^{th}$, that reaches up to 4 km inland. This value is comparable to the 3 km extend reported by Copernicus EMS: "a storm surge also swept 3 km inland, devastating rice paddies and coastal features in the Ebro river delta ..." (at https:// emergency.copernicus.eu/mapping/ems/copernicus-emergency-management-service-monitors-impact-flood-spain). The satellite image indicates that the extension of the flooding caused by the storm was larger than that obtained in our simulation. We explain this apparent discrepancy by the fact that we do not account for the flooding caused by the heavy rains that were reported in the area; instead, our results identify the extent of the flooding caused solely by the marine hazards. Notably, it is worth mentioning that our simulation reproduces the complete flooding of the Trabucador bar, which was swept out by the hazard.

We recall here that the validation of the model outputs indicated that the predicted storm surge in Tarragona tide gauge, the closest station to the delta, was underestimated (Figure 4). Therefore, our simulation could be considered as a lower bound for the flooded area.

## 3.2 Wave impacts in Portocolom (eastern Mallorca Island)

Another location that was severely hit by Storm Gloria was Portocolom, placed in the eastern coast of Mallorca Island (see Figure 1 for the location; top panel Figure 8). This section of the coastline, formed by high cliffs ($\sim 15\,m$), was impacted by large waves that overtopped the cliffs and whose spray reached up to 30 m high (see the inset picture in the low-panel in Figure 8). It caused damages to properties and temporal evacuation of the population.

Our regional simulation provides small storm surge values in the Balearic Islands (Figure 5b), in contrast to the large surges found along the coastline of the mainland. It also provides a maximum $H_s$ of around 4.5 m in the vicinity of Portocolom during

the storm (Figure 5a). The SWASH model has been implemented in 1-D in a section normal to the coast of Portocolom (red line in the upper panel and the insets from Figure 8), as described in the methodology in section 2.2. The initial tests indicated that the modelled $H_s$ of 4.5 m is not high enough as to produce any overtopping. This is not surprising, since modelled $H_s$ underestimates the maximum value recorded by the buoy at Mahon, located in the same area, by an amount of 2.5 m (January 21st at 12:00; red spot in Figure 3); thus, it is likely that the value in Portocolom is also biased low. Therefore, given that there is an amount of visual confirmations that overtopping was produced in this section of the cliffs (see the list in the Appendix, for example), we designed a set of SWASH simulations with values of $H_s$ ranging from 5 m up to 10 m with a step of 0.5 m and fixing the peak period to 13 s (this is the value estimated by our regional simulation). For each value of $H_s$, we run 100 1-D SWASH simulations, of 60 min each, along the same bathymetric profile to account for the inherent variability in wave height. The averaged water flux over the cliffs and the number of waves that overtopped were computed in every run. These results are represented by the green and orange box plots, respectively, in the lower panel of Figure 8. It can be seen that the first set of 100 simulations that shows overtopping corresponds to a $H_s$ of 7 m. This result suggests that our model is underestimating, at least, around 2.5 of $H_s$, in line with the comparison with observations in the buoy from Mahon (Figure 3). Then, as expected, the value of the flux and the number of waves that overtopped the cliffs increases with $H_s$. As an example, when $H_s$ is 8.5 m, the amount of water over the 15m cliffs is $\sim$ 3600 litres per hour and meter. Note that our values do not account for the water input from the spray which, according to observations, was the most damaging to coastal properties, mostly related to the impact of wave water column against house roofs and balconies.

## 4    Discussion and conclusions

Storm Gloria has produced remarkably diverse impacts along the coasts of the Western Mediterranean. These include flooding of low-lying areas, mostly concentrated in the Ebro Delta (see section 3.1), destruction of coastal infrastructures and intense beach erosion due to the impact of waves. The storm surges and wind waves generated by Storm Gloria have been modelled with high resolution hydrodynamic and spectral wave models, both coupled and uncoupled. The outputs allowed to identify and to quantify the different physical mechanism acting on sectors of the coastlines that included the Spanish mainland (with the low-lying Ebro Delta) and the Balearic Islands. Along the coastline of the mainland, the simulated storm surges reached values up to 70 cm acting together with waves of significant wave height ($H_s$) up to 8 m. In contrast, the storm surge was negligible along the coasts of the Balearic Islands, where the reported impacts were mainly attributed to the high waves reaching the shore. Here, we have reproduced the overtopping of these waves over the high cliffs in the eastern coast of Mallorca Island using a non-hydrostatic model forced with the output of the regional wave model. In addition, by using a combination of different numerical simulations, we determined the role of each forcing factor (among atmospheric pressure, wind and ocean waves) in generating the total elevation of the storm surges and we conclude that the coupling between the hydrodynamic model and the spectral wave model is essential to account for the wave setup. Wave setup is particularly relevant in the Ebro Delta, where the model has predicted the flooding induced by the storm surge reaching up to 4 km inland and coincident with

reports of monitoring services. On the other hand, the contribution of the tides to the flooding of the Ebro Delta was neglected
since Storm Gloria lasted enough to include low and high tides with maximum tidal amplitude less than 9 cm.

It is worth noting that Storm Gloria arrived after two relatively recent sea storms of lower intensity and return period of roughly 5 and 10 years, and before the complete recovery of the sedimentary coasts in the region was achieved. The frequency, and not only the intensity of the waves, have been shown to have an important effect on the eroded sediment in Mediterranean beaches during this type of events (Morales-Márquez et al., 2018). For instance, in January 2017 the basin experienced a mistral
sea storm with maximum significant wave height values around 30 h, but in April 2018 there was an unusual sea storm event that maintain continuously maximum significant wave height values over more than 5 days. This sequence has very likely contributed to the large beach erosion induced by Storm Gloria in many places.

A major question arising with respect to the impacts of Storm Gloria is how rare this event has been and how it compares in intensity and extension with past events. Earlier works that have analysed the wave climate over the Western Mediterranean
provide values of $H_s$ that allow putting Storm Gloria into a historical perspective. Cañellas et al. (2007) used the HIPOCAS dataset (Soares et al., 2002), spanning the period 1958-2001, and estimated that 50-year return period in $H_s$ is 11 m in the central part of the basin and has smaller values of 8 m along the eastern coast of the Iberian Peninsula and the southern part of the Balearic Islands. The same data base has shown that extreme wave heights over the Catalan and Valencia coast are significantly lower than those obtained in the north of the Balearic Islands due to the shadow effect of the islands over the
intense north fetch produced by the Gulf of Lions and Ligurian sea storms (Ponce de León and Guedes Soares, 2010). These results indicate that, unlike Storm Gloria, extreme events are in most cases associated to northerly winds. The values extracted for maximum $H_s$ using HIPOCAS data base are listed in Table 1 for two grid points located at the north and south of the Balearic Islands. At the northern point there are at least 11 events with $H_s$ larger than 7 m during a minimum of six hours, being the most energetic the storms of December 1980 and January 2001, that reach respectively 13.9 m and 10.4 m in $H_s$
and lasting for more than 3 days. At the southern point, in contrast, there are at least other 11 storm events with a significantly lower $H_s$. In this case, the most energetic storms correspond to December 1967 and December 1980, with $H_s$ of 7.35 m and 8.93 m, respectively. In other words, the large waves generated by Storm Gloria of the order of 8 m in $H_s$ are found among the largest events over the Western Mediterranean basin. The characteristics that have made Storm Gloria exceptional in terms of coastal impacts are, on the one hand, that while the largest waves are generally found in the northern sector of
the Western Mediterranean, caused by northerly winds, during this event, the largest waves occurred in the western part of the domain and very close to the coast, increasing their potentially hazardous effects. This is further illustrated in Figure S.M. 2a, where maximum $H_s$ for the period 2007-2017 is mapped. This output has been extracted from the Mediterranean Sea Waves Hindcast (CMEMS MED-Waves) available through Copernicus Monitoring Environment Marine Service (CMEMS) (Korres et al., 2019) and is the most recent, high-resolution wave hindcast in the region. Figure S.M. 2b shows the equivalent quantile
of the maximum values of $H_s$ reached during Storm Gloria and indicates that, while there have been other high values in the central and northern parts of the basin for the hindcasted period, the waves generated by the storm in the area between the Balearic Islands and the mainland are not found in these records (quantile values over 0.99). On the other hand, coastal impacts of Storm Gloria are mainly linked to the large storm surges occurred along the coasts of the mainland. Such temporarily high

sea levels, reaching sustained values around half a meter during two days, have exacerbated the coastal hazards in this area to
the extent of causing unprecedented damages (the appendix compiles a non-comprehensive list of reported damages published
through official agencies as well as in the media). According to Kopp et al. (2014), this value of 50 cm corresponds to the
median projected mean sea-level rise in the Mediterranean Sea by year $\sim 2080$ under RCP4.5 and $\sim 2070$ under RCP8.5.
Thus, what is exceptional in the present-day climate, may become the average conditions later this century.

**Appendix**

A non-comprehensive list of websites with reports, images and videos of damages and impacts of Storm Gloria:

1. http://www.aemet.es/es/noticias/2020/01/Tres_temporales_mediterraneos_en_nueve_meses

2. https://beteve.cat/medi-ambient/platges-barcelona-estat-temporal-gloria/

3. http://www.caib.es/pidip2front/jsp/ca/fitxa-convocatoria/strongdesperfectes-i-pegraverdues-materials-als-ports-de-les-illes-balears-a-causa-de-la-borrasca-glograveriastrong

4. https://ib3.org/desallotgen-primera-linia-cala-marcal-mallorca-portocolom-ones-superen-altura-habitatges

5. https://www.diariodemallorca.es/mallorca/2020/01/22/temporal-mallorca-deja-balance-334/1479830.html

6. https://www.diariodemallorca.es/mallorca/2020/01/21/borrasca-gloria-mallorca-mejores-imagenes/1479528.html

7. https://www.20minutos.es/noticia/4126870/0/balance-muertos-danos-desaparecidos-borrasca-gloria-enero-2020/

8. https://lahoradigital.com/noticia/24641/sociedad/la-borrasca-gloria-la-peor-tormenta-de-levante-de-este-siglo.html

9. https://elpais.com/elpais/2020/01/20/album/1579518566_774901.html

10. https://www.elperiodico.com/es/tiempo/20200124/temporal-catalunya-cataluna-borrasca-gloria-ultimas-noticias-directo-7812567

11. https://www.infobae.com/america/mundo/2020/01/21/los-videos-que-reflejan-la-ferocidad-de-la-tormenta-gloria-en-espana-que-acumula-tres-muertes-apagones-y-escuelas-cerradas/

12. https://www.lavanguardia.com/vida/20200123/473082325223/temporal-gloria-balance-muertos-danos-espana-cataluna.html

*Video supplement.* A video of the numerical simulation is available in the supplementary material.

*Author contributions.* AA and MM conceived the work and designed the numerical experiments. DSC retrieved the atmospheric forcings.
AA performed the numerical simulations. AA and MM analysed the outputs and all authors contributed to the outline and writing of the
manuscript.

*Competing interests.* No competing interests are present.

*Acknowledgements.* This study was supported by FEDER/Ministerio de Ciencia, Innovación y Universidades – Agencia Estatal de Investigación through MOCCA project (grant number RTI2018-093941-B-C31). D. S. Carrió was supported by the ARC Centre of Excellence for Climate Extremes (CEI70100023).

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

| North ($H_s > 7m$) | | | | South ($H_s > 6m$) | | | |
|---|---|---|---|---|---|---|---|
| Year | $H_s\,(m)$ | $T_p\,(s)$ | $D_p\,(°)$ | $Dur\,(h)$ | Year | $H_s\,(m)$ | $T_p\,(s)$ | $D_p\,(°)$ | $Dur\,(h)$ |

| North ($H_s > 7m$) | | | | | South ($H_s > 6m$) | | | | |
|---|---|---|---|---|---|---|---|---|---|
| Year | $H_s\,(m)$ | $T_p\,(s)$ | $D_p\,(°)$ | $Dur\,(h)$ | Year | $H_s\,(m)$ | $T_p\,(s)$ | $D_p\,(°)$ | $Dur\,(h)$ |
| 1960 | 7,44 | 11,17 | 22 | 205 | 1960 | 6,45 | 10,15 | 68 | 48 |
| 1967 | 7,46 | 12,28 | 35 | 160 | 1961 | 6,23 | 9,33 | 75 | 64 |
| 1972 | 7,35 | 11,17 | 19 | 39 | 1965 | 6,04 | 11,13 | 83 | 88 |
| 1979 | 7,43 | 11,17 | 19 | 74 | **1967** | **7,35** | **11,21** | **33** | **47** |
| **1980** | **13,87** | **14,86** | **25** | **121** | 1971 | 6,56 | 9,23 | 113 | 69 |
| 1982 | 7,81 | 12,28 | 22 | 69 | 1973 | 7,14 | 11,15 | 106 | 85 |
| 1986 | 7,24 | 9,22 | 291 | 88 | **1980** | **8,93** | **10,71** | **75** | **109** |
| 1987 | 7,12 | 10,12 | 294 | 80 | 1992 | 6,84 | 10,11 | 108 | 83 |
| 1996 | 7,45 | 9,00 | 315 | 113 | 1993 | 6,36 | 10,15 | 103 | 135 |
| 1997 | 7,69 | 10,15 | 25 | 58 | 1997 | 6,89 | 10,21 | 115 | 87 |
| **2001** | **10,44** | **12,38** | **29** | **204** | 2001 | 6,13 | 8,26 | 79 | 85 |

**Table 1.** Characteristics of the events with maximum significant wave height $H_s$ at two grid points north and south of the Balearic Islands, extracted from the HIPOCAS data base (Soares et al., 2002). Strongest events are highlighted in bold.

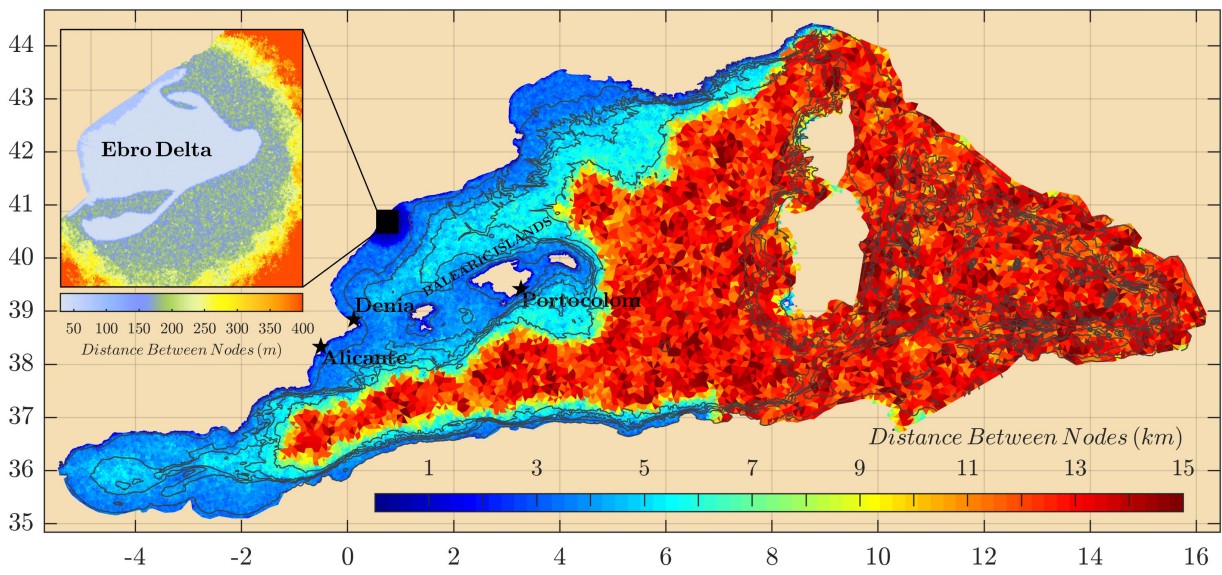

**Figure 1.** Map of the simulation domain with the element size indicated by colours. The contour lines indicate the 3000 m, 2500 m, 2000 m, 1500 m, 1000 m, and 100 m isobaths. The upper-left corner zoom shows the element size over the Ebro Delta river.

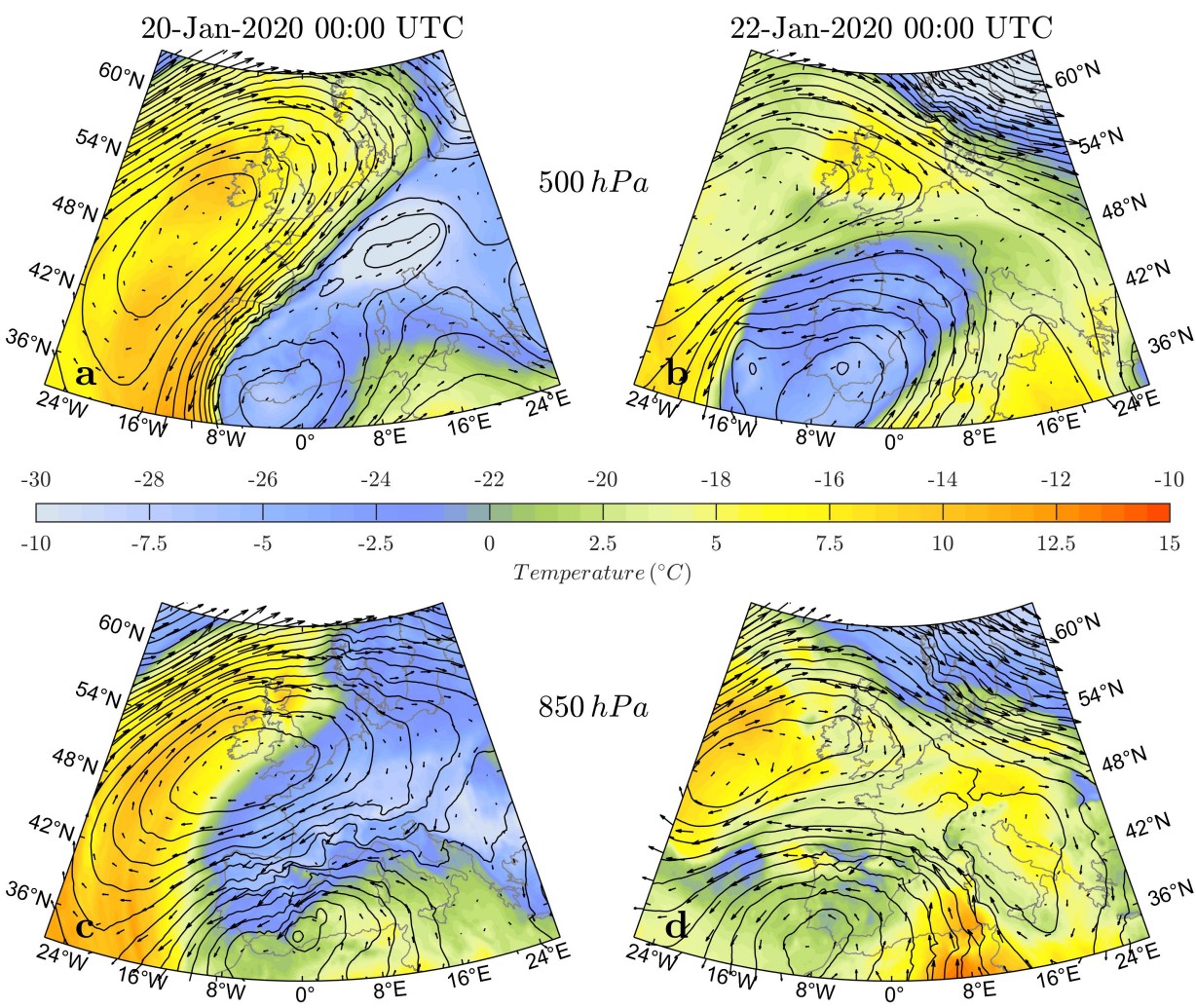

**Figure 2.** ECMWF analyses at 500 hPa (a and b) and 850 hPa (c and d) at 0000 UTC on January $20^{th}$ (a and c) and $22^{nd}$ (b and d) 2020. Black solid lines show the geopotential height, colours indicate the temperature and wind speed is represented by arrows. Note the different temperature ranges for the 500 hPa $[-30°C, -10°C]$ and 850 hPa $[-10°C, 15°C]$ fields.

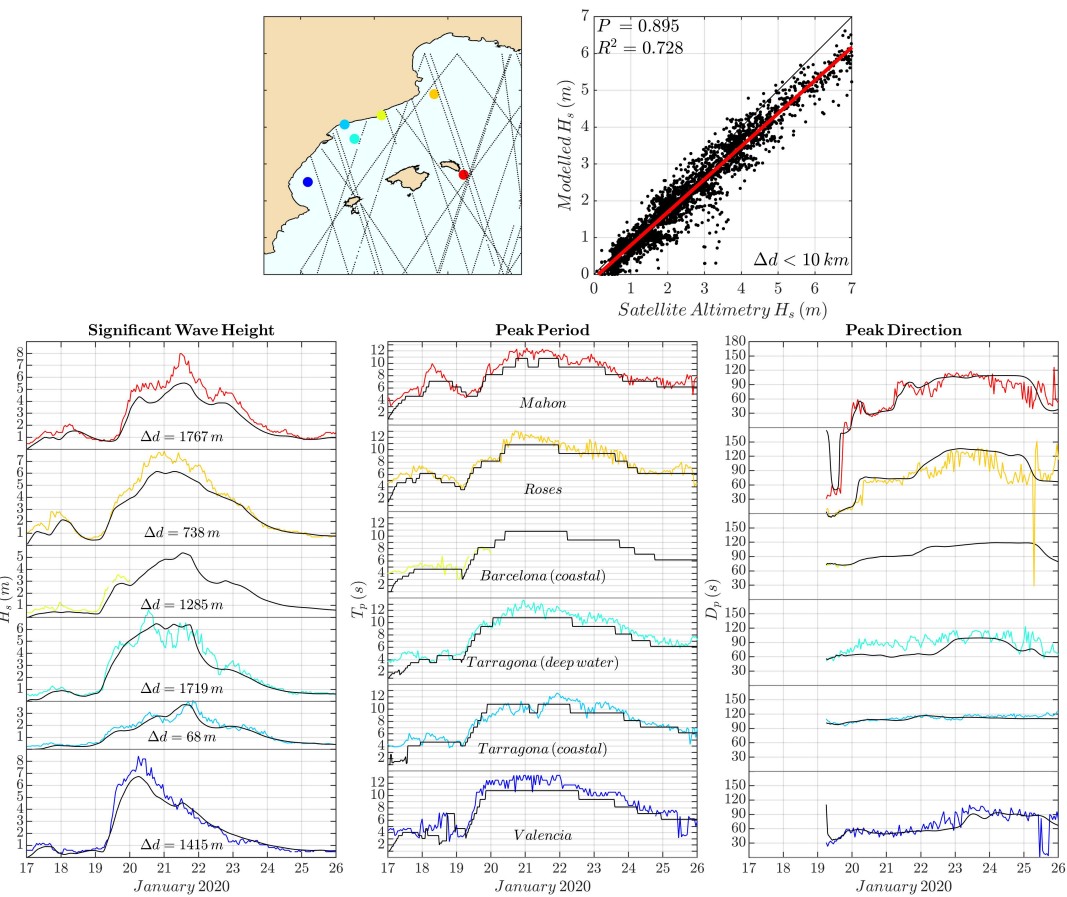

**Figure 3.** Comparison between modelled (black line) and observed (coloured lines; each colour corresponds to one location in the map) wave parameters during the storm. Note that wave direction before 19/01 is not used because of the low confidence with very small waves. The upper right panel shows the comparison between the modelled significant wave height and the satellite-measured at the same location (satellite tracks are indicated by black lines in the map) and time. $\Delta d$ indicates the distance between observations and the closest grid point used for comparison.

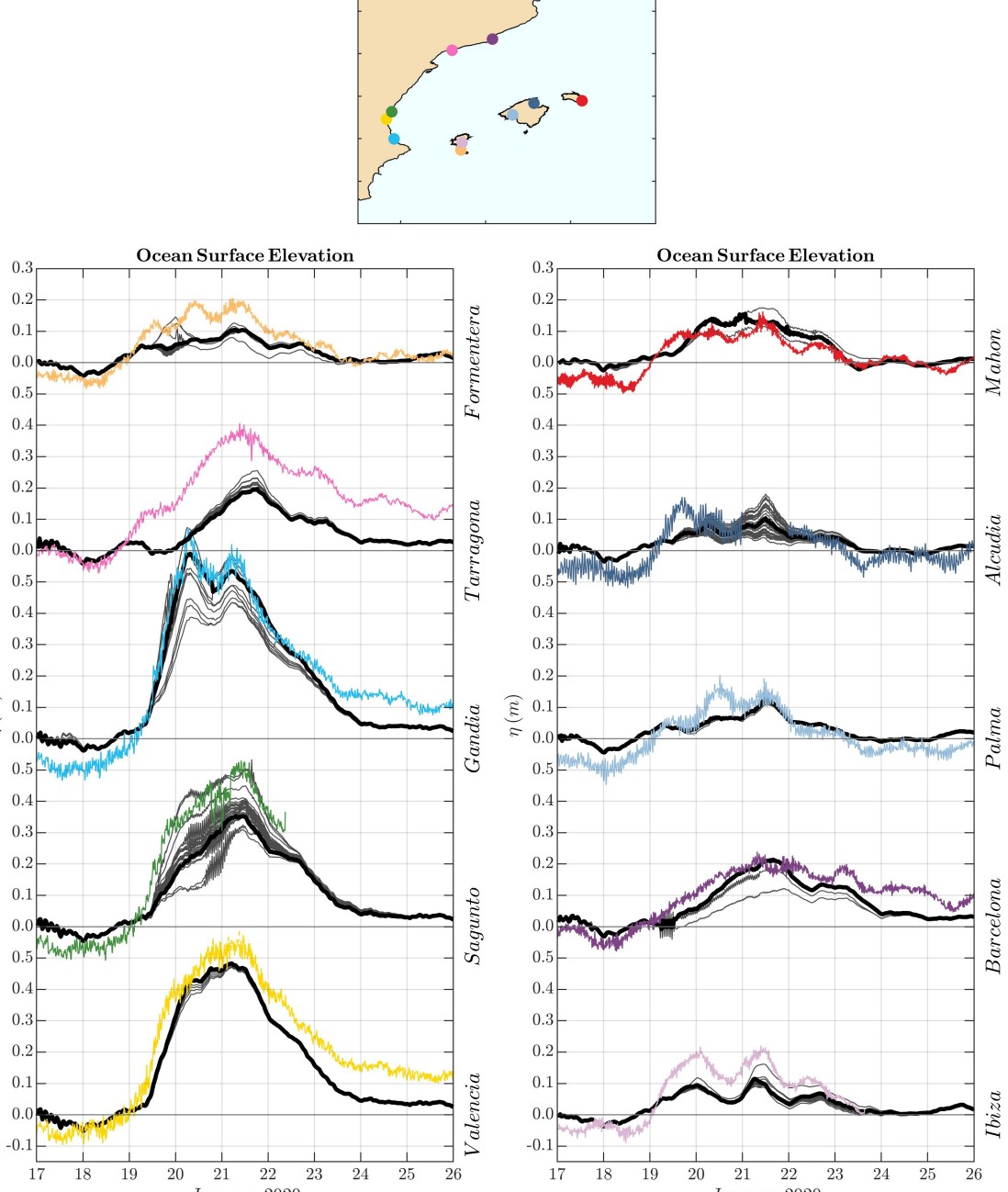

**Figure 4.** Comparison between modelled (black lines) and observed (coloured lines; each colour corresponds to one location in the map) storm surges by tide gauges. To avoid strong bathymetric dependencies, the time series of all modelled points closer to 2.5 km to the tide gauge location are shown. The closest modelled point to the tide gauge is indicated by a thick black line.

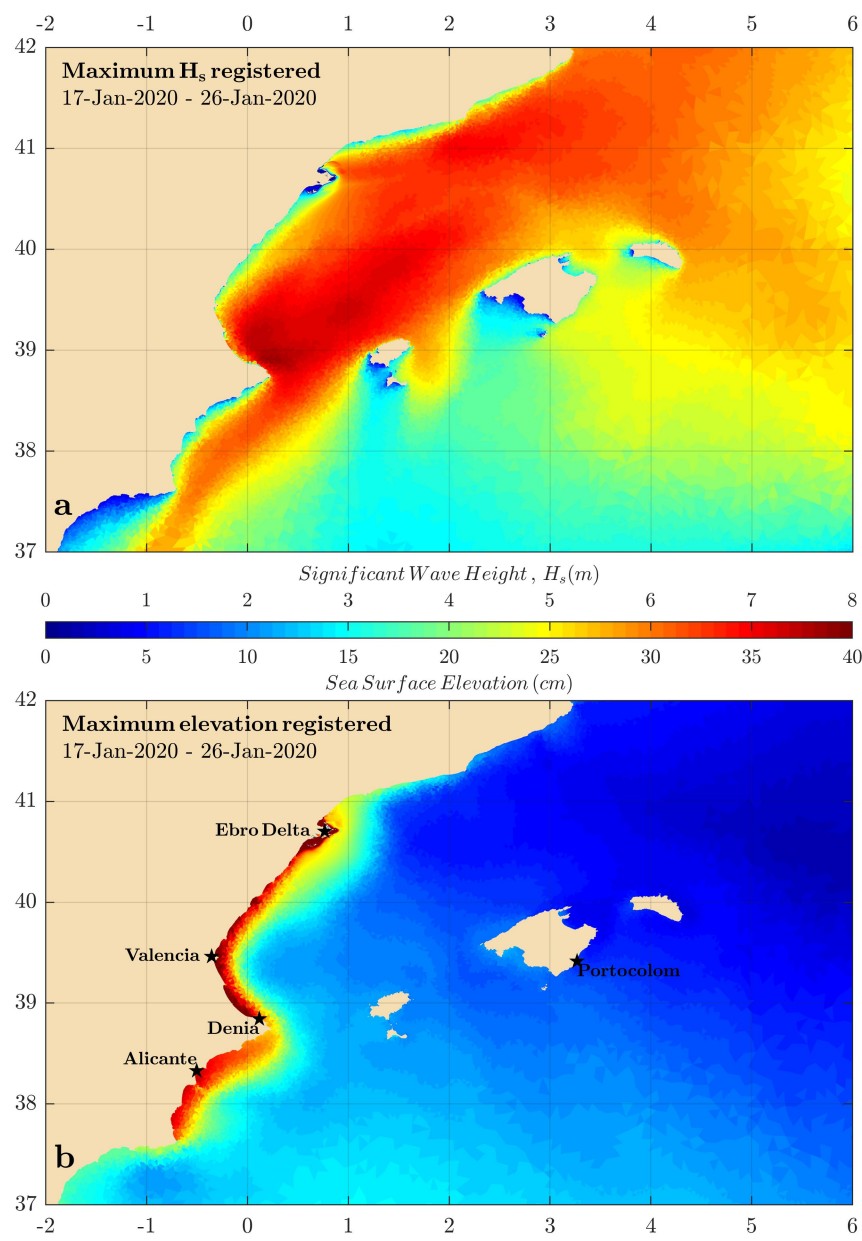

**Figure 5.** Maximum values of $H_s$ (a) and sea surface elevation (b) during the simulation period.

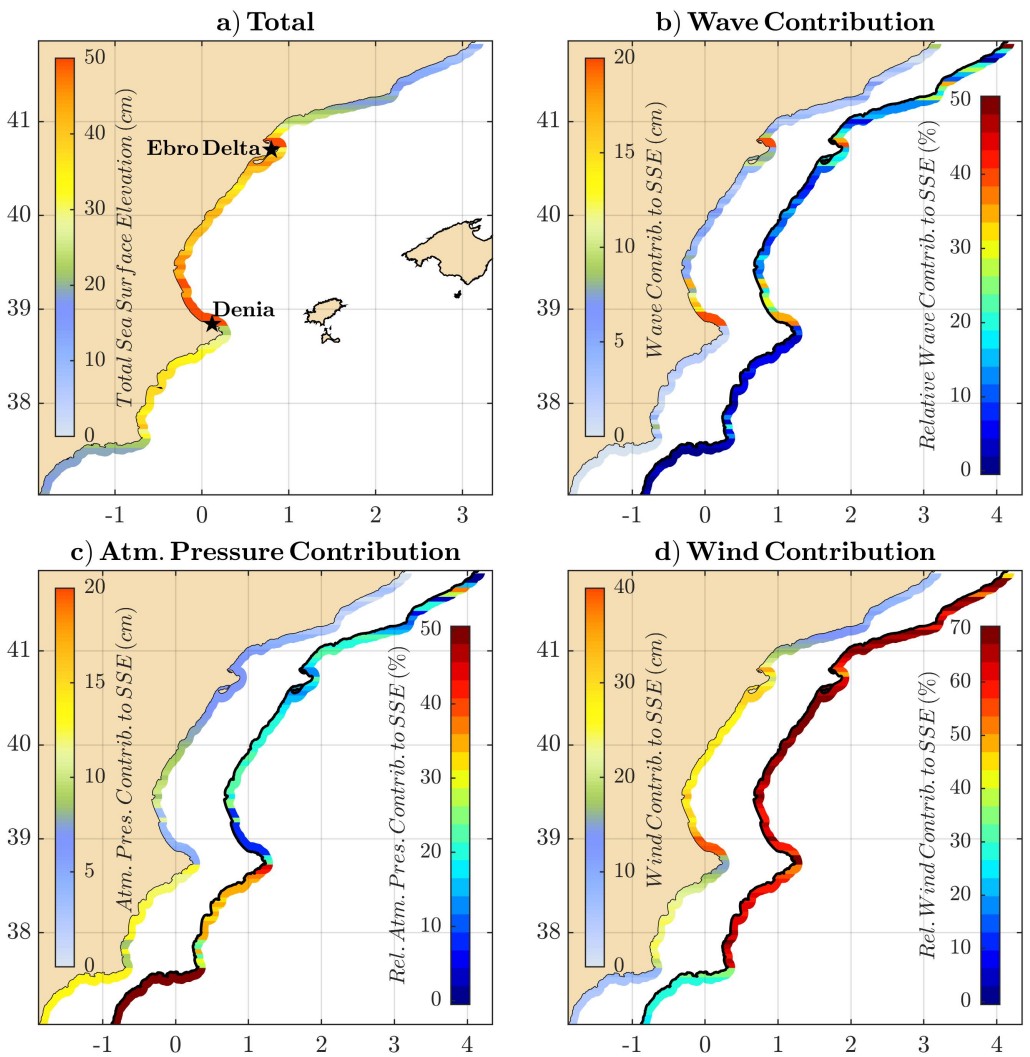

**Figure 6.** Maximum storm surge along a coastal stripe affected by Storm Gloria (a) and the contributors: wave setup (b), atmospheric pressure (c) and wind (d). In panels b, c and d, the absolute (relative) contribution is indicated by the profile in the left (right). Note that the colour scales for the wind have higher limits.

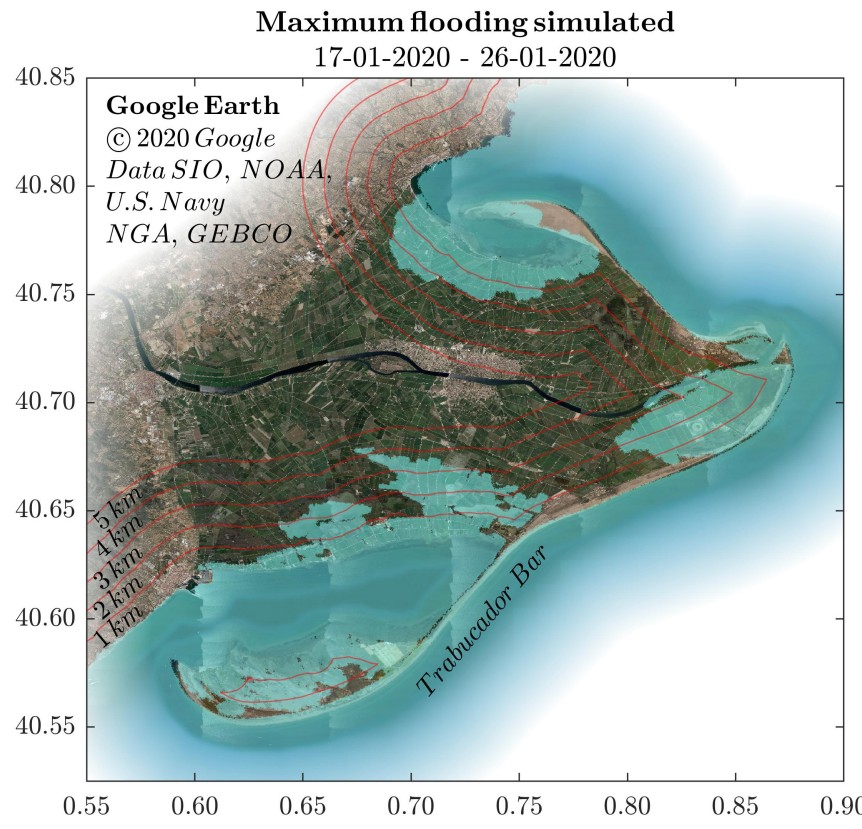

**Figure 7.** Simulated flooded area in the Ebro Delta caused by the storm surge. Red lines indicate distance from coast.

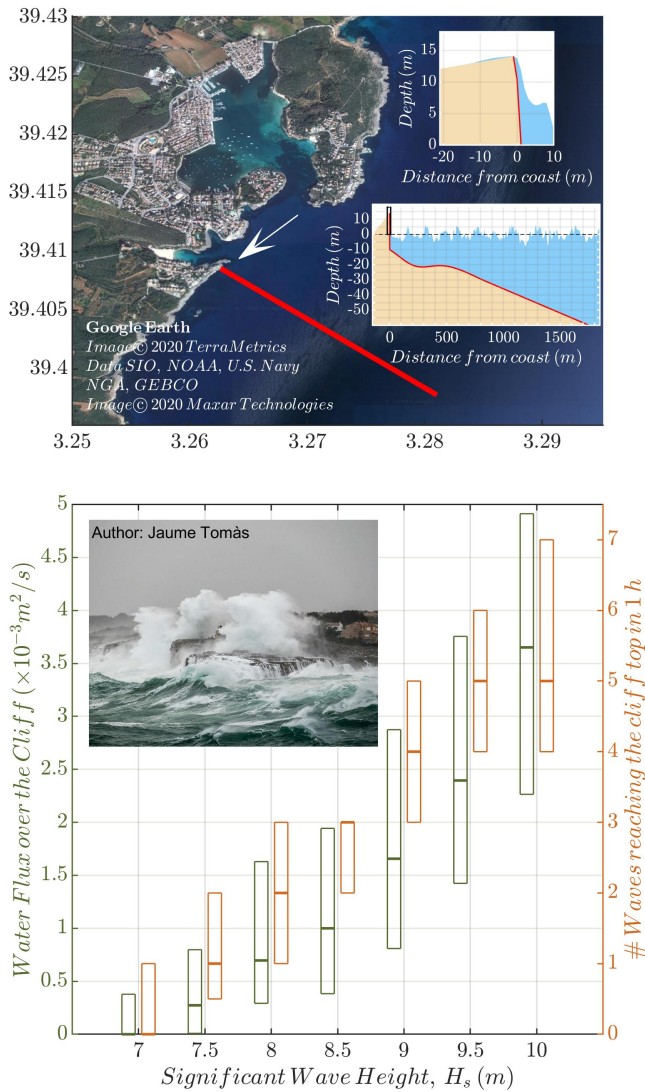

**Figure 8.** Upper panel: Satellite image of Portocolom area with the transect where the bathymetric profile was extracted in red. The lower inset shows one snapshot of one SWASH simulation in one moment where there is overtopping; the upper inset shows a zoom of the upper part of the cliff corresponding to the black box in the lower inset. Bottom panel: box plots of the water flux over the cliff per lineal meter (green boxes) and the number of waves reaching the top of the cliff (dark orange boxes) as a function of significant wave height. The thick line of the box plots indicate the median extracted from 100 different 1D SWASH simulations while the bottom and the top of the boxes show the quantile 0.25 and 0.75, respectively. The inset picture shows the waves hitting the simulated spot (white arrow in the upper panel) during the most intense time of Storm Gloria, Januaty 21st at 12:00 UTC.