# Peer review of "Coastal Impacts of Storm Gloria (January 2020) over the Northwestern Mediterranean"

_Natural Hazards and Earth System Sciences, 2020_

## Referee Comment (RC1) · Anonymous Referee #1 · 21 Apr 2020

The reviewed manuscript "Coastal Impacts of Storm Gloria (January 2020) over the Northwestern Mediterranean" is a numerical study on storm surge, primarily using SCHISM for hydrodynamics and WWM-III for wave dynamics. A baseline 2D model was set up and validated considering the compound effects of wave, atmospheric pressure, and wind. The contribution from each effect were investigated individually by sensitivity tests. Locally high-resolution was implemented in the 2D mesh for a coastal site; a 1D non-hydrostatic model was implemented for another local region with high cliffs using SWASH. The simulation results of Storm Gloria were analyzed and then put into a historical context. The research is the earliest model study on Storm Gloria. The set up and validation of the numerical model are rigorous. The discussion on individual contributors of the total surge, spatial variabilities and historical context are of scientific

and practical importance. I find the manuscript very well written. It generally meets NHESS's standard (attached in the previous page); only minor revisions are required. Specific comments 1) The authors should try to expand on the analysis of the spatially varying wave contributions to the total surge, specifically on why there are two hotspots (Ebro Delta and Denia in Figure 6b) along the coast. In Section 3 (Ln 194), Ebro Delta and Denia are found to differ from other along-shore regions in wave contribution (>20 cm, compared to mostly <7 cm elsewhere; 40-50% of the total surge, compared to mostly <10% elsewhere, as estimated from Figure 6b). Is this pattern related to shore-line geometry, topography/bathymetry, or forcing? Does mesh resolution have anything to do with it (seems not, since Denia is not refined)? Please elaborate either before or within Section 3.1; a short paragraph or 2-4 sentences will do.

2) A short paragraph needs to be added in Section 4, summarizing the major accomplishment and findings of the current work. Right now, the last paragraph (which I assume serves as the conclusion) only slightly touches the current work in the 2nd sentence. Technical corrections 1) Ln 55: consider adding some background for the two selected localities. Did you select them arbitrarily as long as they differ in morphology and forcing? Are they the most severely impacted area? Do they have any significance in agriculture, human residence, or wild life habitat? Some aspects are mentioned later, but a brief description here before delving into the modeling work would be nice.

2) Ln 62: discusses the results and "provides" the final remarks.

3) Ln 86: More details should be provided on the model setup, e.g.: dt, bottom friction, etc. Also consider showing the computation speed, e.g., number/type of cores and the ratio of simulation time to real time.

4) Ln 94: use the multiplication symbol instead of "x".

5) Ln 120: "m" should be in normal font.

[Figure]

6) Ln 121: provide a brief explanation on why a non-hydrostatic model is needed here in addition to the coupled SCHISM-WWMIII model, so that readers with less background can follow.

7) Ln 135: because model results were not interpolated onto observation points, the authors should provide the maximum distance among all pairs of observation and model grid points.

8) Ln 141: Add one or two sentences, providing possible causes of underestimating Hs.

9) Ln 158: "cm" should not be italic.

10) Ln 158: provide possible causes of underestimating elevation at Tarragona. Uncertainties in forcing, DEM, etc.?

11) Ln 197-202: [no corrections needed] If differentiating river flooding and storm surge is of interest to the authors, there are some recent publications on compound flood modeling using SCHISM and WWMIII.

12) Ln 277: . . . a mistral sea storm "with" maximum significant wave height . . .

13) Figure 6: put the subplot labels (a,b,c,d) into the titles.

---

## Referee Comment (RC2) · Anonymous Referee #2 · 5 May 2020

Suitability. The subject of the paper, i.e. the study of the Coastal Impacts of Storm Gloria (January 2020) over the Northwestern Mediterranean falls within the fields covered by NHESS.

Summary. The paper objective it twofold, concentrating on the shorelines of the eastern Spanish coasts and the Balearic Islands: (1) quantify at a regional scale the physical mechanisms at play along the different coastal areas in the basin, including the storm surges and the effect of waves, and discuss their differences, (2) at a more local scale (Ebro Delta and cliff of the eastern Mallorca Island), simulate the impacts of the storm, accounting for the storm surge and wave setup. The paper provides key results on the significant contribution of wind-induced storm surge at the regional scale along the mailand, and wave overtopping at Mallorca cliff site. It also provides flood modeling

results on the Ebro delta.

General comment.While the manuscript is very clear, well written and provides interesting insights in the knowledge of the Gloria storm marine forcings, the manuscript has some weaknesses which deserve to be tackled before publication: discussion (or integration?) on neglected marine forcing especially for the local flood investigation and for the regional model validation (tide, water level fluctuations induced by 3D circulations), the method used to validate the storm surge model, the validity of regional model to properly estimate wave setup with a grid resolution of 1-2km. In addition, the manuscript would benefit from a bit more physical interpretation of the results.

Major remarks

1.Role of neglected processes?

3D Mediterranean circulation induce seasonal water level fluctuations of several centimeters to tens centimeters (Arnicol e al., 1995, Bouffard and Pascuale, 2008). For example, Arnicol et al (1995) indicate variations of +/- 10 cm at the scale of the whole Mediterranean Sea and of each of the two basins. Such fluctuation is far to be negligible for flood issues in micro-tidal areas as the study sites. A bit of discussion on this water level contribution during the Gloria storm would be useful and could reinforce the confidence in the results, if, for instance, this contribution contributed for almost zero during Gloria storm. In addition, all the modeling experiment seem to neglect the tide. The authors should make more explicit that they neglect the tide and discuss the implications on the results for the flood investigations (Ebro Delta and Balearic rocky cliff). Indeed, for instance, the maximum tidal range seems far to be negligible (0.85 m in Barcelona after http://www.portdebarcelona.cat/en/web/el-port/101#2) in front of the Gloria storm surge. But what was the tide during Gloria storm?

2.Model resolution for the wave setup quantification

Without more justification, a coastal resolution of 1-2km is probably too coarse to capture the local wave set-up contribution. Either I am wrong, then the authors should prove that this resolution is enough for their study site. Or I am true, and then, I am afraid that the authors should remove the analysis of the wave setup contribution (at the regional scale). But they could probably discuss it for the Ebro Delta, where the grid resolution falls to 30 meters (and thus is probably fine enough to capture the wave setup).

3.Model validation

First, regarding the wave model results, the manuscript would benefit from explanations for the Hs underestimation. Second, and more importantly, I have some doubts with the method which consists in comparing the water level model outputs in the 5km radius to the local tide gauge measurements. Indeed, depending on the grid points, some points may include a part of the wave setup (probably less than the reality due to the too coarse resolution of the model, except close to the Ebro delta area), others not. As illustrated in Figure 4, there is a strong variability in the model outputs in the 5km radius, which makes not fully convincing the conclusion of a model providing satisfactory prediction compared to the tide gauge measurements. I would suggest at least to add the model outputs of the nearest point to the tide gauge (simulation #1 and #2). In theory (if the grid resolution is high enough to capture the wave setup), the tide gauge measurements should be comprised more or less between the results of simulation #1 and #2, for the nearest point. If there are discrepancies, the authors could discuss the location and resolution of the model close to the tide-gauge (with maps) and also discuss the local knowledge of wave setup contribution to the tide gauge measurements. To contribute to provide a clearer validation and keep the 5km radius, another idea could be to plot all the model outputs for simulations #1 (first) but with a colorscale (on the time series of model outputs) indicating the distance of the model outputs to the tide gauge, and put in thick line the closest point (together with the tide gauge observation of course). The same figure could be done with the simulations #2 (together with the tide gauge observation of course). Of course, the authors are

free to follow other ideas, as long as it makes the validation clearer by at least showing results on the closest point. I think this an important issue. Refining the validation could also help identifying to which extend the seasonal water level fluctuations induced by 3D circulations are negligible or not during the Gloria storm.

"On-line" Remarks

- Title: for me, the main focus of the paper is not on providing information on coastal impacts, but more on investigating the relative forcing contributions. I would suggest to modify the title to better illustrate the paper content.

- Abstract: The abstract could be a bit more informative regarding the key results.

- Line 38: please provide the geographical coordinates of the Mahon buoy.

- Line 72: Figure 4 is called before Figure 3 -> reorder the figures?

- Line 85: "contains" -> "contain"

- Line 104-107: Test 3 & 4 are done with the 2DH hydrodynamic model or with the coupled model? If the first case, the authors should make it more explicit, and then in Lines 107-108 stress that these tests 3 and 4 are used to estimate the contribution of Patm and wind on the atmospheric storm surge.

- Line 110-112: not clear if the 0% and 3% come from theoretical analysis or from the modeling results. Please clarify.

- Line 121-132: it seems that steady forcing conditions (for SWASH) are used in terms of wave spectrum and still water level. More justification/explanation of this choice and its implication wuld be wellcome.

- Line 129: "an initial integration time of 0.05 s" -> "an initial computation time step of 0.05 s".

- Line 131: "0,5 m" -> "0.5 m"

- Line 149: explain/justify why the tide gauge data have been low-pass filtered using a Butterworth filter with a cutoff period of 30 minutes. I guess this is due to some local physical reasons, but some justification would be welcome.

- Line 161: add a subsection title?

- Line 164: not sure the authors can use "ocean" for the Mediterranean Sea -> reformulate?

- Line 174-175 / "This pattern is caused by the winds blowing towards the mainland": I do not fully agree. Indeed, for me, the results are also strongly influenced by the bathymetry. I remind that the analyss of the 2DH shallow water equations show that wind-induced storm surges increase with decreasing water depth (see e.g. Flather (2001)). I think the authors could easily check it using their simulation results (by having a look on 2D spatial maps of simulation #4). This remark leads also to the suggestion to add a bathymetric map in the paper. This will support the analysis of the forcing contributions.

- Line 207-210: these sentences are not clear to me. Please clarify.

- Line 204-214: the comments on the validation/comparison of the model results in terms of flood are not really clear to me. Indeed, when I compare the Copernicus map and the model results, the model seems to provide a larger flooded surface, but predict no flood in one of the N-E area, while there was flood. This this is not clear to me why the authors seem to think that the model underestimates the flood. The manuscript would probably benefit from quantitatively comparing the Copernicus map and the model results, for instance with a map showing the following classes: Copernicus and model predict no flood; Copernicus and model predict flood; Copernicus indicates flood, but the model predicts no flood; Copernicus indicates no flood, but the model predict flood ; Copernicus and the model predict no flood. If not accessible, the Copernicus map could be digitized. In addition, at the Ebro scale, this could be interesting and relevant to investigate the spatial variations relative contribution of wave

set-up, pressure induced and wind induced storm surges (more in details that in figure 6).

- Legend of Figure 2: "c and c" should be "a and c"?

———————————————————

References

- Arnicol G., Le Traon P.Y., Ayoub N., De Mey P. (1995) Mean sea level and surface circulation variability of the Mediterranean Sea from 2 years of TOPEX/POSEIDON altimetry. J. of Geophys. Res., 100(C12), 25, p. 163-25, 177.

- Bouffard J and Pascual A. (2008) A review of altimetry Applications over European Coasts (invited talk). Second Coastal Altimetry Workshop, Pisa, Italy, 2008.

- Flather, R. A.: Storm surges, in: Encyclopedia of Ocean Sciences, edited by: Steele, J. H., Thorpe, S. A., and Turekian, K. K., Academic, San Diego, Calif, 2882–2892, 2001.

---

## Short Comment (SC1) · 21 May 2020

Please note we encountered problems uploading the revised version. We have already contacted the editorial office to solve it.

---

## Short Comment (SC2) · 21 May 2020

Please note we encountered problems uploading the revised version. We have already contacted the editorial office to solve it.

---

## Author Comment (AC1) · 21 May 2020

**Response to Referee #1 of our manuscript entitled**
**Coastal Impacts of Storm Gloria (January 2020) over the Northwestern Mediterranean**
**[nhess-2020-75] submitted to *Natural Hazards and Earth System Sciences.**

Angel Amores, Marta Marcos, Diego S. Carrió, and Lluís Gómez-Pujol

May 21, 2020

**Author's response:** We would like to thank the Reviewer for the comments provided. We have responded point by point to all the concerns raised below, with indication of the changes in the manuscript:

The reviewed manuscript "Coastal Impacts of Storm Gloria (January 2020) over the Northwestern Mediterranean" is a numerical study on storm surge, primarily using SCHISM for hydrodynamics and WWM-III for wave dynamics. A baseline 2D model was set up and validated considering the compound effects of wave, atmospheric pressure, and wind. The contribution from each effect were investigated individually by sensitivity tests. Locally high-resolution was implemented in the 2D mesh for a coastal site; a 1D non-hydrostatic model was implemented for another local region with high cliffs using SWASH. The simulation results of Storm Gloria were analyzed and then put into a historical context. The research is the earliest model study on Storm Gloria. The set up and validation of the numerical model are rigorous. The discussion on individual contributors of the total surge, spatial variabilities and historical context are of scientific and practical importance. I find the manuscript very well written. It generally meets NHESS's standard (attached in the previous page); only minor revisions are required.

Specific comments:
1) The authors should try to expand on the analysis of the spatially varying wave contributions to the total surge, specifically on why there are two hotspots (Ebro Delta and Denia in Figure 6b) along the coast. In Section 3 (Ln 194), Ebro Delta and Denia are found to differ from other alongshore regions in wave contribution ($> 20$ cm, compared to mostly $< 7$ cm elsewhere; 40-50% of the total surge, compared to mostly ¡10% elsewhere, as estimated from Figure 6b). Is this pattern related to shoreline geometry, topography/bathymetry, or forcing? Does mesh resolution have anything to do with it (seems not, since Denia is not refined)? Please elaborate either before or within Section 3.1; a short paragraph or 2-4 sentences will do.

**Author's response:** To illustrate our response we have produced the Figure below, that maps significant wave height ($H_s$, top panel) and wave peak direction ($D_p$, bottom panel) at the time that Storm Gloria hit stronger along the coast of the mainland (January $20^{th}$, 2020). As the reviewer states the maxima wind-wave contributions to the total surge in Denia and the Northern side of the Ebro Delta is a physical effect linked to the wave direction. It is not related to the grid resolution since, as the reviewer noted, the area around Denia is not refined. These two spots were the areas where the waves hit the coast more perpendicularly and, consequently, the wave setup was larger. We thus conclude that the observed pattern in these two spots is a combination of the forcing (with large $H_s$ and that direction) and the shoreline geometry, coinciding with the direction perpendicular to the forcing. We have added a paragraph explaining this fact just before section 3.1, following reviewer's advice.

[Figure]

2) A short paragraph needs to be added in Section 4, summarizing the major accomplishment and findings of the current work. Right now, the last paragraph (which I assume serves as the conclusion) only slightly touches the current work in the 2nd sentence.

**Author's response:** We have re-arranged the last section, now including a paragraph where we summarise the main findings of the present study.

Technical corrections:
1) Ln 55: consider adding some background for the two selected localities. Did you select them arbitrarily as long as they differ in morphology and forcing? Are they the most severely impacted area? Do they have any significance in agriculture, human residence, or wild life habitat? Some aspects are mentioned later, but a brief description here before delving into the modeling work would be nice.

**Author's response:** We selected these two locations based on a combination of two factors: differences in morphology and in the forcing, as stated in the text. In addition, for the local studies we needed high resolution topo-bathymetries to perform the local studies, that are not available everywhere but they were for these two areas.
We have included some background of these two spots in the introduction (second-to-last paragraph).

2) Ln 62: discusses the results and "provides" the final remarks.

**Author's response:** This change has been introduced.

3) Ln 86: More details should be provided on the model setup, e.g.: dt, bottom friction, etc. Also consider showing the computation speed, e.g., number/type of cores and the ratio of simulation time to real time.

**Author's response:** We have included more information about the model setup in the first paragraph of section 2.2

4) Ln 94: use the multiplication symbol instead of "x".

**Author's response:** This change has been introduced.

5) Ln 120: "m" should be in normal font.

**Author's response:** This change has been introduced.

6) Ln 121: provide a brief explanation on why a non-hydrostatic model is needed here in addition to the coupled SCHISM-WWMIII model, so that readers with less background can follow.

**Author's response:** We have included a sentence in lines 129-131 explaining the reason why a non-hydrostatic model is needed at this point (last paragraph of section 2.2).

7) Ln 135: because model results were not interpolated onto observation points, the authors should provide the maximum distance among all pairs of observation and model grid points.

**Author's response:** The distance between the location of the buoys and the closest model grid point has now been included in the Fig. 3 as insets in the panel of the $H_s$ ($\Delta d =$...). The values range between 68 m and 1.7 km. This is referenced at the beginning of section 2.3.

8) Ln 141: Add one or two sentences, providing possible causes of underestimating Hs.

**Author's response:** Possible causes are a poor quality of the atmospheric forcing, a bad performance of the numerical model or inaccurate bathymetry. To test the model performance, we repeated the simulation with the SWAN wave model and obtained the same outputs, so this cause can be discarded. The atmospheric forcing slightly underestimates the wind during the peak of the storm (see Figure 1 in S.M), which might have an effect together with the possibly limited representation of the bathymetry. We have now included a brief discussion of these possible causes in section 2.3 (second paragraph).

9) Ln 158: "cm" should not be italic.

**Author's response:** This change has been introduced.

10) Ln 158: provide possible causes of underestimating elevation at Tarragona. Uncertainties in forcing, DEM, etc.?

**Author's response:** We believe that when approaching the coast the major source of error is the bathymetry, which is likely not accurate enough. We have included a sentence in this respect at the end of section 2.3.

11) Ln 197-202: [no corrections needed] If differentiating river flooding and storm surge is of interest to the authors, there are some recent publications on compound flood modeling using SCHISM and WWMIII.

**Author's response:** Thanks to the reviewer for the heads up. We will check these publications.

12) Ln 277: . . . a mistral sea storm "with" maximum significant wave height . . .

**Author's response:** This change has been introduced.

13) Figure 6: put the subplot labels (a,b,c,d) into the titles.

**Author's response:** This change has been introduced.

---

## Author Comment (AC2) · 21 May 2020

**Response to Referee #2 of our manuscript entitled**
**Coastal Impacts of Storm Gloria (January 2020) over the Northwestern Mediterranean**

[nhess-2020-75] submitted to *Natural Hazards and Earth System Sciences.*

Angel Amores, Marta Marcos, Diego S. Carrió, and Lluís Gómez-Pujol

May 21, 2020

**Author's response:** We would like to thank the Reviewer for the assessment on our manuscript and for the comments provided. We have responded to all the concerns raised below, with indication of the changes in the manuscript:

Suitability. The subject of the paper, i.e. the study of the Coastal Impacts of Storm Gloria (January 2020) over the Northwestern Mediterranean falls within the fields covered by NHESS.

Summary. The paper objective it twofold, concentrating on the shorelines of the eastern Spanish coasts and the Balearic Islands: (1) quantify at a regional scale the physical mechanisms at play along the different coastal areas in the basin, including the storm surges and the effect of waves, and discuss their differences, (2) at a more local scale (Ebro Delta and cliff of the eastern Mallorca Island), simulate the impacts of the storm, accounting for the storm surge and wave setup. The paper provides key results on the significant contribution of wind-induced storm surge at the regional scale along the mainland, and wave overtopping at Mallorca cliff site. It also provides flood modeling results on the Ebro delta. General comment. While the manuscript is very clear, well written and provides interesting insights in the knowledge of the Gloria storm marine forcings, the manuscript has some weaknesses which deserve to be tackled before publication: discussion (or integration?) on neglected marine forcing especially for the local flood investigation and for the regional model validation (tide, water level fluctuations induced by 3D circulations), the method used to validate the storm surge model, the validity of regional model to properly estimate wave setup with a grid resolution of 1-2km. In addition, the manuscript would benefit from a bit more physical interpretation of the results.

**Major remarks:**

1.Role of neglected processes? 3D Mediterranean circulation induce seasonal water level fluctuations of several centimeters to tens centimeters [Bouffard and Pascual; Larnicol et al., 1995]. For example, Larnicol et al. [1995] indicate variations of +/- 10 cm at the scale of the whole Mediterranean Sea and of each of the two basins. Such fluctuation is far to be negligible for flood issues in micro-tidal areas as the study sites. A bit of discussion on this water level contribution during the Gloria storm would be useful and could reinforce the confidence in the results, if, for instance, this contribution contributed for almost zero during Gloria storm.

**Author's response:** Seasonal sea level changes in the Mediterranean Sea range between 4 and 8 cm for the annual and 2 and 4 cm for the semi-annual signals, on average [*Marcos and Tsimplis*, 2007], in agreement with the magnitude pointed out by the reviewer. We have explored this baroclinic signal as illustrated in the figure below, corresponding to the tide gauge record in Barcelona. The time series has been demeaned and detrended. According to *Marcos and Tsimplis* [2007], in the western Mediterranean basin the sea level seasonal cycle peaks between September-November, and decreases afterwards. This is observed in the figure (lower panel). This panel also shows that mean sea level during Storm Gloria varies around the zero, which corresponds to the averaged mean sea level during the tide gauge period. Therefore, we conclude that seasonal mean sea level does not add any further effect that amplified or reduced the impacts of the storm.

[Figure]

Figure 1: Sea level time series at Barcelona tide gauge (upper panel) and zoom of the most recent period (lower panel)

In addition, all the modeling experiment seem to neglect the tide. The authors should make more explicit that they neglect the tide and discuss the implications on the results for the flood investigations (Ebro Delta and Balearic rocky cliff). Indeed, for instance, the maximum tidal range seems far to be negligible (0.85 m in Barcelona after `http://www.portdebarcelona.cat/en/web/el-port/101#2`) in front of the Gloria storm surge. But what was the tide during Gloria storm?

**Author's response:** There seems to be an error in the tidal range at the site referenced by the reviewer. The tides in Barcelona, as in most of the Mediterranean Sea are much smaller. This is observed in the figure above of the previous response. Also, please check

the table below, that has been extracted from the website of the Spanish Port Authority for the same tide gauge. It lists the tidal harmonics in Barcelona computed for the time period 1993-2018 (`http://www.puertos.es/en-us/oceanografia/Pages/portus.aspx`). The largest tidal constituent is the M2 with 4.60 cm of amplitude. Summation of all tidal constituents results in a total amplitude of 17.28 cm what makes a maximum tidal range of 34.56 cm, value that is far from the 85 cm stated by the authorities of the Barcelona harbour in which seems to be a typo.

In our simulation we did not consider the tides for the reason outlined above and also because the storm lasted 3 days and included all tidal phases. We have now specified this in the text. We have added a sentence in the manuscript indicating that the tides are not taken into account in the simulation (2nd paragraph in section 2.2).

**Armónicos de Marea calculados sobre el periodo 1993-2018** / *1993-2018 Harmonic Constituents*

| Armónico Harmonic Id | Frecuencia Frequency (ciclos/hora) | Amplitud Amplitude (cm) | Fase Phase (°) | Armónico Harmonic Id | Frecuencia Frequency (ciclos/hora) | Amplitud Amplitude (cm) | Fase Phase (°) |
|---|---|---|---|---|---|---|---|
| Z0 | 0.000000 | 30.05 | 0.00 | S2 | 0.083333 | 1.65 | 230.58 |
| Q1 | 0.037219 | 0.32 | 53.01 | K2 | 0.083561 | 0.48 | 228.54 |
| O1 | 0.038731 | 2.36 | 102.80 | M3 | 0.120767 | 0.14 | 158.85 |
| P1 | 0.041553 | 1.25 | 160.81 | MN4 | 0.159511 | 0.21 | 303.21 |
| K1 | 0.041781 | 3.68 | 168.03 | M4 | 0.161023 | 0.52 | 346.81 |
| 2N2 | 0.077487 | 0.15 | 190.35 | SN4 | 0.162333 | 0.05 | 4.38 |
| MU2 | 0.077689 | 0.16 | 177.26 | MS4 | 0.163845 | 0.33 | 51.23 |
| N2 | 0.078999 | 0.98 | 201.44 | MK4 | 0.164073 | 0.10 | 58.78 |
| NU2 | 0.079202 | 0.18 | 202.60 | | | | |
| M2 | 0.080511 | 4.60 | 213.38 | | | | |
| L2 | 0.082024 | 0.12 | 220.89 | | | | |

Data extracted from: http://www.puertos.es/es-es/oceanografia/Paginas/portus.aspx

**Tabla generada por Puertos del Estado**/*Generated by* **Puertos del Estado**          **Fecha actual**/*Today is* **05 May 2020**

2.Model resolution for the wave setup quantification Without more justification, a coastal resolution of 1-2km is probably too coarse to capture the local wave set-up contribution. Either I am wrong, then the authors should prove that this resolution is enough for their study site. Or I am true, and then, I am afraid that the authors should remove the analysis of the wave setup contribution (at the regional scale). But they could probably discuss it for the Ebro Delta, where the grid resolution falls to 30 meters (and thus is probably fine enough to capture the wave setup).

**Author's response:** We would like to highlight that the comparison between modelled and observed sea level as measured by tide gauges (Figure 4 in the manuscript) shows a good agreement, suggesting that our simulation correctly captures the most relevant processes that are taking place during Storm Gloria. In order to prove this, we are going to focus our response in the two sites where the wave setup has significantly contributed most to the total water level modelled along the Mediterranean coast of the Iberian Peninsula, i.e. the northern side of the Ebro Delta and the region around Denia (see Figure 6 of the Manuscript). Since the high grid resolution of the Ebro Delta could explain by itself the good agreement, we are discussing here the results and comparison between model and observations at Gandia tide gauge which is the closest one to Denia (Figure 4). In the figure below we show the same comparison as in Figure 4 for Gandia tide gauge, but we

have added the ocean surface elevation time series of the simulation without taking into account the waves (red lines) to the coupled simulation (grey lines). Not accounting for the wave setup (red lines) underestimates by around 20 cm the observed sea level (blue line), whereas including the effect of waves (grey lines) decreases this bias. Indeed, the closest grid point (thick grey line) mimics the amplitude of the observed storm surge. We thus conclude that the spatial resolution that we are using is enough to, at least partially, capture the effects of the wave setup.

[Figure]

3.Model validation. First, regarding the wave model results, the manuscript would benefit from explanations for the Hs underestimation.

**Author's response:** This was also a query from Reviewer #1. Possible causes are a poor quality of the atmospheric forcing, a bad performance of the numerical model or inaccurate bathymetry. To test the model performance, we repeated the simulation with the SWAN wave model and obtained the same outputs, so this cause can be discarded. The atmospheric forcing slightly underestimates the wind during the peak of the storm (see Figure 1 in S.M), which might have an effect together with the possibly limited representation of the bathymetry.

We have added a discussion on the possible causes of the underestimation of $H_s$ (section 2.3, 2nd paragraph).

Second, and more importantly, I have some doubts with the method which consists in comparing the water level model outputs in the 5km radius to the local tide gauge measurements. Indeed, depending on the grid points, some points may include a part of the wave setup (probably less

than the reality due to the too coarse resolution of the model, except close to the Ebro delta area), others not. As illustrated in Figure 4, there is a strong variability in the model outputs in the 5km radius, which makes not fully convincing the conclusion of a model providing satisfactory prediction compared to the tide gauge measurements. I would suggest at least to add the model outputs of the nearest point to the tide gauge (simulation #1 and #2). In theory (if the grid resolution is high enough to capture the wave setup), the tide gauge measurements should be comprised more or less between the results of simulation #1 and #2, for the nearest point. If there are discrepancies, the authors could discuss the location and resolution of the model close to the tide-gauge (with maps) and also discuss the local knowledge of wave setup contribution to the tide gauge measurements. To contribute to provide a clearer validation and keep the 5km radius, another idea could be to plot all the model outputs for simulations #1 (first) but with a colorscale (on the time series of model outputs) indicating the distance of the model outputs to the tide gauge, and put in thick line the closest point (together with the tide gauge observation of course). The same figure could be done with the simulations #2 (together with the tide gauge observation of course). Of course, the authors are free to follow other ideas, as long as it makes the validation clearer by at least showing results on the closest point. I think this an important issue. Refining the validation could also help identifying to which extend the seasonal water level fluctuations induced by 3D circulations are negligible or not during the Gloria storm.

**Author's response:** We have explored different options to meet the reviewer requirements and we decided that the best one is to represent the time series of the modelled points within a radius of 2.5 km to the tide gauge location and indicating the closest grid point with a thicker line (see the new Figure 4).

We also produced alternative figures following reviewer's recommendations, including a different color lines depending of the distance for grid point (see the figure below). This format is, in our opinion, less clear and hinders its interpretation. In addition, we also produced a figure merging the results of simulations #1 and #2 (not shown here) but again it looked too messy. We hope that the reviewer's concern about the ability of the model in resolving the wave setup was satisfied with the answer to the question #2. The results showed essentially the same conclusions as in our example for Gandia tide gauge discussed above.

We hope that the new Figure 4 is more satisfactory for the reviewer.

[Figure]

**"On-line" Remarks:**
- Title: for me, the main focus of the paper is not on providing information on coastal impacts, but more on investigating the relative forcing contributions. I would suggest to modify the title to better illustrate the paper content.

**Author's response:** We have carefully considered reviewer's criticism regarding the title of the manuscript. We understand the reluctance to focus on the impacts, since we interpret that the reviewer associates the term "impacts" to only our two case studies. However, to our view, coastal impacts refer to the effects that the storm had on the physical mechanisms acting along the coast and that included the storm surges and waves. In this sense, and this was our initial purpose, we want to highlight the marine impacts along the coasts of the Western Mediterranean of Storm Gloria. We have been trying to figure out a not too long title that accounts for regional as well as local effects, but without success (our best approach is the first sentence of the abstract). Among all the alternatives, our preferred is the current title and we would like to keep it as it is.

- Abstract: The abstract could be a bit more informative regarding the key results.

**Author's response:** We are limited here by the maximum allowed length of the abstract, between 100 and 200 words. We are currently using 209 words which makes it impossible to extend without removing some of the major results that are highlighted.

- Line 38: please provide the geographical coordinates of the Mahon buoy.

**Author's response:** The coordinates have been introduced in the text.

- Line 72: Figure 4 is called before Figure 3 − > reorder the figures?

**Author's response:** Thanks for spotting this error. We have decided to switch the order of the text instead of the figures (change in 2nd and 3rd paragraph in section 2.3)

- Line 85: "contains" − > "contain"

**Author's response:** The whole sentence is: *This small region covering the Delta and its surroundings contains around 75% of the grid nodes.*; as the subject is "This small region", the verb should be contains.

- Line 104-107: Test 3 & 4 are done with the 2DH hydrodynamic model or with the coupled model? If the first case, the authors should make it more explicit, and then in Lines 107-108 stress that these tests 3 and 4 are used to estimate the contribution of Patm and wind on the atmospheric storm surge.

**Author's response:** The simulations #3 and #4 are done with the 2DH hydrodynamic model (i.e. uncoupled). It is written in lines 112-113 that the simulation #3 is "a hydrodynamic model run forced only by atmospheric pressure" and that the simulation #4 is " a hydrodynamic model run forced only by wind". Moreover, in lines 115-116 it is indicated that "the contribution of the atmospheric pressure (wind) was determined with the run #3 (#4)".

- Line 110-112: not clear if the 0% and 3% come from theoretical analysis or from the modeling results. Please clarify.

**Author's response:** This values come from the modelling results. We have modified the text in line 119 to clarify this issue.

- Line 121-132: it seems that steady forcing conditions (for SWASH) are used in terms of wave spectrum and still water level. More justification/explanation of this choice and its implication would be welcome.

**Author's response:** We have used the steady conditions because we aim at determining the minimum significant wave height needed to have overtopping on the cliffs. So we have used a range of values of $H_s$. This is now explained in the manuscript (last paragraph, section 2.2).

- Line 129: "an initial integration time of 0.05 s" − > "an initial computation time step of 0.05 s".

**Author's response:** This change has been introduced.

- Line 131: "0,5 m" − > "0.5 m"

**Author's response:** This change has been introduced.

- Line 149: explain/justify why the tide gauge data have been low-pass filtered using a Butterworth filter with a cutoff period of 30 minutes. I guess this is due to some local physical reasons, but some justification would be welcome.

**Author's response:** The reviewer is right. We low-pass filtered the time series of the tide gauges to remove the signal of the resonant modes of the harbours which are usually less than 30 minutes. We have included a sentence clarifying the reason of this filtering (last paragraph, section 2.3).

- Line 161: add a subsection title?

**Author's response:** We did not add a subsection on purpose. We first describe the regional results of the model runs and then use subsections only for the two case studies.

- Line 164: not sure the authors can use "ocean" for the Mediterranean Sea − > reformulate?

**Author's response:** In this case we use ocean as a synonym of sea, since it is used very closed to the word sea: "...together with the ocean responses in sea surface elevation".

- Line 174-175 / "This pattern is caused by the winds blowing towards the mainland": I do not fully agree. Indeed, for me, the results are also strongly influenced by the bathymetry. I remind that the analysis of the 2DH shallow water equations show that wind-induced storm surges increase with decreasing water depth (see e.g. Flather [2001]). I think the authors could easily check it using their simulation results (by having a look on 2D spatial maps of simulation #4). This remark leads also to the suggestion to add a bathymetric map in the paper. This will support the analysis of the forcing contributions.

**Author's response:** We agree with the reviewer. The shallow waters along the coasts of the mainland play a role in the magnitude of the storm surges. We have added a line in this respect to clarify the sentence (2nd paragraph, section 3). We have also followed reviewer's recommendation and we have modified Figure 1 to include bathymetric lines that will facilitate the interpretation of the new sentence.

- Line 207-210: these sentences are not clear to me. Please clarify.

**Author's response:** We have rewritten the sentence, that now reads:"The satellite image indicates that the extension of the flooding caused by the storm was larger than that obtained in our simulation. We explain this apparent discrepancy by the fact that we do not account for the flooding caused by the heavy rains that were reported in the area; instead, our results identify the extent of the flooding caused solely by the marine hazards."

- Line 204-214: the comments on the validation/comparison of the model results in terms of flood are not really clear to me. Indeed, when I compare the Copernicus map and the model results, the model seems to provide a larger flooded surface, but predict no flood in one of the N-E area, while there was flood. This this is not clear to me why the authors seem to think that the model underestimates the flood. The manuscript would probably benefit from quantitatively comparing the Copernicus map and the model results, for instance with a map showing the following classes: Copernicus and model predict no flood; Copernicus and model predict flood; Copernicus indicates flood, but the model predicts no flood; Copernicus indicates no flood, but the model predict flood ; Copernicus and the model predict no flood. If not accessible, the Copernicus map could be digitized. In addition, at the Ebro scale, this could be interesting and relevant to investigate the spatial variations relative contribution of wave set-up, pressure induced and wind induced storm surges (more in details that in figure 6).

**Author's response:** We would like to remark that, according to the reports, part of the flooding in the Ebro Delta during Storm Gloria was caused by heavy rains. These are not included in our simulation and therefore a direct quantitative comparison would not make sense. Our purpose here was to estimate as accurately as possible the marine-induced flooding. We added the comparison to the satellite image to argue that the extension of the modelled flooding was realistic. In this respect, we feel that we have achieved our goal with the map we have represented in Figure 7 that shows the areas that, for sure, were flooded by salty water.

- Legend of Figure 2: "c and c" should be "a and c"?

**Author's response:** This change has been introduced.

**References**

Bouffard, J., and A. Pascual (), A review of altimetry Applications over European Coasts (invited talk), *Second Coastal Altimetry Workshop, Pisa, Italy.*

Flather, R. A. (2001), Storm surges, *Encyclopedia of Ocean Sciences, edited by: Steele, J. H., Thorpe, S. A., and Turekian, K. K.*, p. 2882–2892.

Larnicol, G., P.-Y. Le Traon, N. Ayoub, and P. De Mey (1995), Mean sea level and surface circulation variability of the Mediterranean Sea from 2 years of TOPEX/POSEIDON altimetry, *Journal of Geophysical Research: Oceans*, *100*(C12), 25,163–25,177, doi:10. 1029/95JC01961.

Marcos, M., and M. N. Tsimplis (2007), Variations of the seasonal sea level cycle in southern Europe, *Journal of Geophysical Research: Oceans*, *112*(C12), doi:10.1029/2006JC004049.

---

## Author Response (AR2)

**Response to Referee #2 of our manuscript entitled**
**Coastal Impacts of Storm Gloria (January 2020) over the Northwestern Mediterranean**
[nhess-2020-75] submitted to *Natural Hazards and Earth System Sciences.* Second revision.

Angel Amores, Marta Marcos, Diego S. Carrió, and Lluís Gómez-Pujol

June 3, 2020

**Author's response:** We would like to thank the Reviewer for the assessment on our manuscript and for the comments provided. We have responded to all the concerns raised below, with indication of the changes in the manuscript:

The authors answered and took into account most of my comments. However, there are still remaining questions/corrections.

A/ Role of neglected processes? Regarding the seasonal sea level close to 0 for Gloria I suggest the authors to provide this information in the manuscript.

**Author's response:** We have added a sentence in section 2.3 (last paragraph) stating explicitly that comparisons between observations and model outputs are referred to mean sea level.

Regarding the tide, tidal range of several tens of centimeters is not negligible when focusing on the flood itself. What was the tidal range for the Gloria event? I can understand one neglect the tide when focusing on nearshore water levels, but I still disagree with the authors when they say that tide is neglected because it is a micro-tidal environment. Indeed, water level changes of more than 10 cm can have a non-negligible effect on the coastal flood, depending on the study site (topography). Even if I understand the argument that the model results are in accordance (qualitative) with the observations for the Ebro Delta flood, I strongly recommend to take into account the tide for the Ebro Delta simulations, or at least to discuss the potential tide contribution to the Ebro Delta flood in the discussion.

**Author's response:** During Storm Gloria, the maximum tidal amplitude was less than 9 cm (maximum tidal range of 19 cm). Its contribution to the flooding of the Ebro Delta was neglected since Storm Gloria lasted enough to include low and high tides. This is now included in the discussion, as suggested by the reviewer (end of first paragraph of the discussion).

B/ Model resolution for the wave setup quantification Regarding the model resolution of the wave set-up, the authors provide a convincing answer. However, no modification has been done in the manuscript. I think that saying in few words what the authors have explained in their answer would reinforce the confidence in the modeling results. I recommend the authors to do this minor correction.

> **Author's response:** We have added a paragraph to the end of section 2.3 explaining our response to this comment and including the same figure as in our previous answer in the supplementary information (now Figure S.M. 3).

C/ Regarding my initial comment on the comparison of flood results of the Ebro Delta, the authors did not completely answer to my remark which whose the first part was : "Line 204-214: the comments on the validation/comparison of the model results in terms of flood are not really clear to me. Indeed, when I compare the Copernicus map and the model results, the model seems to provide a larger flooded surface, but predict no flood in one of the N-E area, while there was flood. This this is not clear to me why the authors seem to think that the model underestimates the flood." That is why also, initially, I suggested to show the Copernicus map in the manuscript.

> **Author's response:** We believe that there might be a confusion regarding the image from Copernicus that we are comparing. At the beginning of section 3.1 we refer to the following website: `http://floodlist.com/europe/spain-storm-gloria-floods-january-2020`. The image here shows how the flooded area covers most of the Ebro Delta, although, as pointed out in the manuscript, it does not discriminate between rain and marine waters. Looking at this map, we clearly have a smaller flooded area. Then, in the second paragraph of the same section, we refer to another Copernicus EMS website: `https://emergency.copernicus.eu/mapping/ems/copernicus-emergency-management-service-monitors-impact-flood-spain`. The image shown here corresponds to January $26^{th}$, that is after the storm, and it does not represent the maximum extend of the flooding. We have referenced this website, though, because it contains information regarding the impacts that we quote in the text.

D/ In the revised manuscript, the authors introduced SWASH as a non-hydrostatic model. This is true that it is a non-hydrostatic model. However, not all non-hydrostatic models are resolving the individual waves, as is doing SWASH. I suggest referring to SWASH as a phase resolving wave model (in comparison with spectral wave model).

> **Author's response:** We have added this term.

[revised manuscript text omitted]